# Changes in Heart Rate, Heart Rate Variability, Breathing Rate, and Skin Temperature throughout Pregnancy and the Impact of Emotions—A Longitudinal Evaluation Using a Sensor Bracelet

**DOI:** 10.3390/s23146620

**Published:** 2023-07-23

**Authors:** Verena Bossung, Adrian Singer, Tiara Ratz, Martina Rothenbühler, Brigitte Leeners, Nina Kimmich

**Affiliations:** 1Department of Obstetrics, University Hospital Zurich (USZ), University of Zurich (UZH), 8091 Zurich, Switzerlandnina.kimmich@usz.ch (N.K.); 2Department of Reproductive Endocrinology, University Hospital Zurich (USZ), University of Zurich (UZH), 8091 Zurich, Switzerland; 3Ava AG, 8001 Zurich, Switzerland

**Keywords:** pregnancy, prenatal care, wearable, sensor, remote monitoring

## Abstract

(1) Background: Basic vital signs change during normal pregnancy as they reflect the adaptation of maternal physiology. Electronic wearables like fitness bracelets have the potential to provide vital signs continuously in the home environment of pregnant women. (2) Methods: We performed a prospective observational study from November 2019 to November 2020 including healthy pregnant women, who recorded their wrist skin temperature, heart rate, heart rate variability, and breathing rate using an electronic wearable. In addition, eight emotions were assessed weekly using five-point Likert scales. Descriptive statistics and a multivariate model were applied to correlate the physiological parameters with maternal emotions. (3) Results: We analyzed data from 23 women using the electronic wearable during pregnancy. We calculated standard curves for each physiological parameter, which partially differed from the literature. We showed a significant association of several emotions like feeling stressed, tired, or happy with the course of physiological parameters. (4) Conclusions: Our data indicate that electronic wearables are helpful for closely observing vital signs in pregnancy and to establish modern curves for the physiological course of these parameters. In addition to physiological adaptation mechanisms and pregnancy disorders, emotions have the potential to influence the course of physiological parameters in pregnancy.

## 1. Introduction

Women show substantial changes in basic physiology during pregnancy, for example, in the cardiovascular and pulmonary systems or in the regulation of body temperature. Most of these changes are physiological for a normal pregnancy, whereas some might be due to an abnormal course or are an indicator of upcoming pregnancy complications. Due to continuous vasodilation, skin temperature rises with the ongoing time of pregnancy [1,2]. Cardiac output is 30–40% higher in the first trimester of pregnancy than in a non-pregnant state. This leads mainly to a higher stroke volume but also to a 15% increase in the heart rate [3]. Besides the abovementioned parameters, various studies show that heart rate variability (HRV), perfusion, and breathing rate are correlated with hormonal levels of progesterone and estrogen [4,5,6]. As both of these hormones strongly increase after conception [7,8], it is assumed that they can influence physiological parameters. Furthermore, emotions can influence maternal vital signs in the course of pregnancy [9,10]. On the other hand, studies have used vital signs in pregnancy to predict emotional changes or depression [11,12].

With our study, we intend to evaluate the change in heart rate, heart rate variability, breathing rate, and skin temperature throughout pregnancy by using an electronic wearable device to prove the concept in close to continuous assessment of these parameters during pregnancy in a home environment and to create standard curves. With the latter, deviations from these standard curves have the potential to reveal upcoming pregnancy complications and might allow for early interventions in the future. In addition, we intend to evaluate the influence of maternal emotions on physiological parameters in pregnancy.

## 2. Related Works and Contributions

Depending on the history of a pregnant woman, pregnancy check-ups are currently performed in intervals of 2 to 5 weeks. As part of these check-ups, physiological parameters, such as heart rate, blood pressure, urine parameters, body temperature, weight, and breathing status are evaluated to detect aberrations from a physiological course. Such an evaluation once every few weeks hides the danger of missing ongoing changes as early as they occur. A detection system with continuous, or at least more frequent, measurements with easy handling for the user in an outpatient setting could help early detection of pregnancy complications, especially in between regular pregnancy check-ups. Electronic wearables, such as widely used fitness bracelets, have the potential to provide such information. These devices are already used for measuring heart rate, oxygen saturation, or body temperature, and to predict ovulation [13,14,15,16].

The AVA sensor bracelet used in our trial has been studied in different settings previously. In a population of 57 healthy women, Zhu et al. compared daily point basal temperature measurements to continuously measured WST using a sensor bracelet in the detection of ovulation [17]. They found WST measured with the sensor bracelet to be more sensitive in these women. Goodale et al. used the AVA bracelet to measure heart rate, HRV, WST, respiratory rate, and skin perfusion in 237 conception-seeking women to study changes during the menstrual cycle [14]. They were able to detect significant alterations in the physiological parameters throughout the cycle using the sensor bracelet. Another project used the AVA bracelet to identify pre-symptomatic COVID-19 cases in a large cohort from Liechtenstein (n = 1163) [18]. The participants wore sensor bracelets during the night and recorded heart rate, respiratory rate, HRV, and WST. The results showed that 68% of the 127 COVID-19 cases could be detected two days before the start of symptoms. Furthermore, the AVA bracelet was used in a small cohort of pregnant women with preterm premature rupture of membranes to detect early signs of intraamniotic infection [19]. Women who developed an infection could be detected through significant differences in breathing rate and heart rate.

Wearable sensors have been used in prenatal care by other research groups. A Finnish cohort of 20 nulliparous pregnant women was monitored continuously using a smart wristband which recorded physical activity, sleep, and heart rate during pregnancy [20]. The authors found the wristband to be feasible for use during pregnancy. Atzmon et al. used a wearable, wireless, noninvasive medical monitor to sample cardiac output, blood pressure, stroke volume, systemic vascular resistance, and heart rate during pregnancy and labor in 81 women [21]. The study focused on the course of these hemodynamic parameters during labor. Grym et al. evaluated the Garmin Vívosmart HR smart wristband to monitor steps, calories, heart rate, stairs climbed, intensity of physical activity, and total hours of sleep, sleep levels, and sleep movement in a cohort of 20 pregnant women [22].

We know from the literature that physiological parameters change during pregnancy, but measurements have rarely been performed on a daily basis over a longer course of pregnancy and, if so, mostly with one-point measurements. Hence, modern curves for the physiological course of these parameters with a high frequency of or even continuous measurements are rarely available during pregnancy. Published studies investigating changes in physiological parameters during pregnancy vary in sample sizes [23,24,25], between ten and several hundred participants, and mostly do not use continuous measurement.

The sensor bracelet used in this study has been proven to be feasible in other cohorts. It has a European CE certification and has been introduced to the market for fertility tracking. The bracelet has the potential to improve prenatal monitoring outside of the clinical setting and to create modern standard curves of physiological parameters during pregnancy using wearable sensors.

## 3. Materials and Methods

### 3.1. Sample Population

This prospective observational study was performed from November 2019 to November 2020. We included healthy pregnant women who met the following inclusion criteria: (a) age 18 to 48 years, (b) gestational age below 32 weeks at inclusion, (c) willing to comply with the study protocol throughout pregnancy, (d) pregnancy care mainly or solely in our institution, (e) sufficient German or English skills, and (f) currently living in Switzerland. A woman was excluded if she (a) had problems wearing the bracelet, (b) had difficulty understanding the study procedures, (c) was taking any medication or other substances that could affect any physiological parameters being studied, (d) was working night shifts or frequently traveling between different time zones, or (e) had a sleeping disorder before pregnancy. We chose the upper limit of 32 weeks for gestational age at inclusion to be able to continuously monitor basic physiological parameters over a longer course of time during pregnancy, as we intended to create modern standard curves using an electronic sensor bracelet.

### 3.2. Data Acquisition

Maternal, fetal, and obstetrical data were extracted from our in-house institutional obstetric database (Perinat version 7.0.0.43).

During the study period, participants measured different physiological parameters continuously during night rest using the Ava Fertility Tracker bracelet (version 2.0, Ava AG, Zurich, Switzerland). Participants received all study materials including detailed instructions about the use of the device. They were instructed to start all study measurements from the first day of enrollment and wore the Ava Fertility Tracker bracelet on the dorsal side of their wrist (always of the same arm) each night while sleeping. The bracelet measured several different physiological parameters simultaneously, including wrist skin temperature (WST), heart rate, heart rate variability, and breathing rate. At least four hours of relatively uninterrupted sleep each night were required for the physiological parameters to stabilize according to the manufacturer’s instructions. The bracelet automatically saved the measurements every 10 s throughout the night. For this study, the first 90 and the last 30 min of each night’s data were excluded to avoid disturbances of the falling-asleep and wake-up phases. Furthermore, the sensor bracelet includes a sleep pacer. All measurements during nighttime, when the woman was not sleeping, were not recorded. For the analysis, the bracelet needs at least four hours of sleep excluding falling asleep and waking up. Temperature data were smoothed using locally weighted scatterplot smoothing (LOESS). The 99th percentile (stable maxima) was chosen out of several percentiles (the 10th, 50th, 90th percentiles) as the daily wrist skin temperature in final analyses [15].

Heart rate is the number of heart beats per minute. Heart rate variability represents the time variation in the interval between two consecutive heart beats. It can be calculated and analyzed in different ways. For this study, the standard deviation of the N-N intervals (SDNN) was used. N-N stands for “normal” beats; abnormal or false beats, like ectopic beats not coming from the sinoatrial node, were removed from the analysis [26].

During the initial interview, participants were shown how to synchronize the device with the complementary Ava app on their smartphones and were instructed to synchronize their data each morning after waking up. During synchronizing, the anonymized device data were transferred to the server. The study participants were asked to wear the bracelet throughout the whole pregnancy. After the study completion, research staff retrieved the bracelet data obtained during the study period from the server and the obstetrical data from the obstetrical database for the final analysis.

Eight emotions (anxious, stressed, tired, sensitive, unmotivated, calm, energized, and happy) were assessed weekly using five-point Likert scales ranging from never to always within the app.

We chose the Ava sensor bracelet for our study as it is non-invasive, discrete, and convenient to wear. Furthermore, it is a CE-approved medical device in Europe and has been proven feasible in other clinical settings. Continuous measurements are more robust than punctual measurements and are potentially able to detect pathologies at an earlier stage in pregnancy. The concept of measuring physiological parameters during sleep excludes various potential interfering factors.

### 3.3. Statistical Analyses

The baseline characteristics of the participants were summarized using descriptive statistics. Continuous parameters were summarized as mean ± SD and categorical parameters as frequency and proportion (%). The weekly mean of the four selected physiological parameters was aggregated per gestational week. The primary outcome was analyzed using linear mixed models, setting the four physiological parameters as dependent variables. To analyze changes in the patterns throughout pregnancy, gestational age in weeks was set as the time variable.

For the statistical analyses, the eight emotions, which had been assessed weekly using five-point Likert scales ranging from never to always within the app, were dichotomized as “most of the time to always” and “sometimes to never”. Due to low cell counts in the frequent feeling of anxiety, it was dichotomous as “sometimes to always” and “rarely to never”. In a first step, bivariate models were run with the eight emotions as predictors. For negative emotions, the category “sometimes to never” was set as reference, whereas for positive emotions, the category “most of the time to always” was set as reference. Finally, all emotions were included in a multivariate model for each physiological parameter, respectively. All models were run by testing the fit of linear versus quadratic time functions, inclusion of a random slope, and inclusion of a time*emotion interaction term, and the model with the best fit was selected using Akaike’s information criterion.

The statistical analyses included all women with available data. No exclusion was performed based on a defined proportion of collected physiological outcome data. All statistical analyses were performed using R (version 4.1.1) via RStudio. All hypotheses were two-tailed. To account for multiple testing and a subsequently increasing family-wise alpha-error, a Bonferroni correction was applied, leading to a *p*-value < 0.001 being considered statistically significant. The models were tested for multicollinearity (based on correlation and variance inflation factor), indicating that none of the emotion indicators had to be removed from the multivariate model due to multicollinearity. However, the emotion “energized” was only included in the bivariate evaluation but excluded from the multivariate models because its inclusion would have reduced the sample size by 50%.

### 3.4. Ethics

The study was carried out in accordance with the Declaration of Helsinki and was approved by the Local Ethical Board (BASEC-No. 2016-02241). The study was registered in the ClinicalTrials.gov database (identifier NCT03161873). Furthermore, all participants gave written informed consent before any study procedures were performed.

## 4. Results

### 4.1. Cohort Characteristics

Thirty-two pregnant women gave their written informed consent for participation in the study, of whom one woman had to be excluded because she had an implanted cardiac device, and eight withdrew their participation for personal reasons without using the wearable at all. Finally, 23 of 32 women (72%) remained for analysis (see Figure 1). The mean age of the cohort was 34.3 years (SD 4.0), and the mean BMI was 23.3 kg/m^2^ (SD 3.4). Most women were nulliparous (17/23), and the mean gestational age at the first measurement with the device was 15.6 weeks (range: 8 to 25 weeks). After inclusion, women wore the bracelet up to 167 nights during the study period. When evaluating the number of days when women wore the bracelet in relation to all the days during the study period when they could have worn the bracelet, the range of compliance during pregnancy was between 6 and 95%.

### 4.2. Heart Rate

Mean resting heart rate measurements ranged from 61.47/min (±0.12/min, gestational week 8) to 67.12/min (±10.57/min, gestational week 35). Looking at the trend line (see Figure 2a), there was a tendency for the resting heart rate to increase with a maximum around 66/min from 24 to 32 weeks. Starting around gestational week 32, it dropped towards the end of pregnancy. Bivariate models suggested an association between low resting heart rate and frequently feeling stressed, as well as an association between high resting heart rate and rarely feeling energized (see Appendix A). The multivariate model for feeling stressed, however, did not show any statistically significant association (see Figure 3a). The emotion “energized” was excluded from the multivariate models because its inclusion would have reduced the sample size by 50% (see methods section for details).

### 4.3. Heart Rate Variability

Weekly mean HRV ranged between 48.17 ms (±4.80 ms, gestational week 12) and 67.95 ms (±5.08 ms, gestational week 8). As opposed to heart rate and breathing rate, it presented a U-shaped pattern, i.e., declining during two-thirds of pregnancy and then starting to rise again, approaching baseline values (see Figure 2b). Frequent feelings of stress were related to an increased HRV, whereas rare feelings of happiness were related to a decreased HRV in bivariate models (see Appendix A Appendix A). Only the association with stress remained statistically significant in multivariate models, with a positive association between HRV and frequent feelings of stress, but a negative association with the time interaction term (see Figure 3b).

### 4.4. Breathing Rate

The weekly mean breathing rate ranged from 12.34/min (±0.15/min, gestational week 10) to 16.18/min (±2.46/min, gestational week 17). The lowest breathing rates of 12 to 13/min were found in early gestation (gestational weeks 9 and 10), increasing up to 16/min around week 17 of pregnancy, with a subsequent decline approaching baseline values towards the end of pregnancy (see Figure 2c). While only the emotion “rarely feeling energized” was associated with a higher breathing rate in the bivariate model (see Appendix A Appendix A), the multivariate model revealed a negative association of the breathing rate with frequently feeling tired, yet there was a positive association with the interaction of tiredness over the course of pregnancy. There also was a positive association with rarely feeling happy in the multivariate analysis, which in turn was a negative interaction with the gestational week (see Figure 3c).

### 4.5. Wrist Skin Temperature

Weekly mean WST presented the lowest variability over time, ranging from 35.23 °C (±0.92 °C, gestational week 37) to 36.54 °C (±0.33 °C, gestational week 10) (see Figure 2d). It was also the only physiological parameter for which the model with a linear function was chosen over the model with a quadratic function. The frequent feeling of anxiety was the only emotion associated with decreased WST in the bivariate model, which remained statistically significant in the multivariate model (see Appendix A and Figure 3d).

## 5. Discussion

We present data from a pilot study using an electronic wearable to collect continuous data on the four physiological parameters: heart rate, breathing rate, heart rate variability, and wrist skin temperature during pregnancy. The vital signs were correlated with eight emotions using bivariate and multivariate analyses, which showed that emotions like feeling happy, stressed, or tired could have an impact on the course of physiological parameters in pregnancy.

Relevant physiological changes occur in the cardiovascular system during pregnancy through the influence of hormonal and hemodynamic changes. The heart rate rises as a response to a decreasing systemic vascular tone [27]. According to published data, it increases by 10 to 20 beats/minute or up to 20% starting at 5–7 gestational weeks, continuing until the end of pregnancy [28,29,30,31,32,33,34,35]. In normotensive pregnancies, a heart rate of 75–83/min was found at under 14 weeks of pregnancy, 79/min at 15–21 weeks, 76–85 at 22–28 weeks, 80–87/min at 29–35 weeks, 84–90/min at 36–41 weeks, and 69/min at 3–6 months postpartum [24,29,34,36]. Heart rate during nighttime is known to be lower than during daytime, with rates of 68/min during the night in early pregnancy and 73/min in late pregnancy compared to 88/min and 96/min during the day, respectively [29]. The heart rate reported here was measured during nighttime but does not represent the mean nightly heart rate, because the 10th percentile of the nightly measured heart rate was chosen to aggregate the parameter. This might explain why the nightly heart rate was lower in our study compared to previously reported values. We did observe a slight increase in heart rate throughout gestation, which corroborates earlier findings. We did not find any statistically significant association between heart rate and the studied emotions. An abnormal increase in maternal heart rate can be caused by different pathologies like anemia, hyperthyroidism, or heart disease and can be detected by continuous measurement with a wearable sensor during pregnancy.

Heart rate variability is a marker of the autonomic control of the heart and represents beat-to-beat variation in the cardiac interval [37]. According to previous studies, it decreases over the course of pregnancy, which seems to represent an increasing influence of sympathetic activity towards the end of pregnancy compared to a more parasympathetic tone at the beginning [29,37,38,39,40]. Additionally, it is influenced by physical activity, with an increase in heart rate variability and parasympathetic tone during exercise [41]. We observed a decrease in HRV during the first two-thirds of pregnancy as expected based on prior research findings. Furthermore, we observed a slight increase, approaching baseline values, towards the end of pregnancy, which has been described by others who used a smartwatch for monitoring in pregnancy [42]. We found the decrease in HRV in pregnancy over time to be more pronounced in women who frequently felt stressed, which is in line with previously reported findings [9]. The continuous monitoring of HRV during pregnancy could be used as an indicator for upcoming hypertensive disorders in pregnancy [43,44] as well as maternal mental well-being [11,45].

The respiratory tract function changes during pregnancy, induced by hormonal and anatomical changes. The thoracic circumference increases, the diaphragm is elevated, chest wall compliance decreases by the influence of pregnancy hormones, and the uterus expands within the abdomen [46]. In addition, most functional lung parameters change, such as an increase in oxygen consumption and ventilation [28,46,47]. However, according to the literature, breathing rate remains more or less stable throughout the course of pregnancy with a rate of 14–19 breaths per minute [28,47,48]. The findings from our study match previous reports of stable breathing rates throughout pregnancy, with slightly declining breathing rate around 16 to 14.5 breaths per minute from the second trimester onwards; yet, we observed a slight increase in early pregnancy. Although one expects breathing rates to be stable throughout pregnancy, we observed that a slight decrease in the number of breaths per night over time was pronounced in women who rarely felt happy, whereas a slight increase seemed to be more pronounced in women who frequently felt tired. Since regression coefficients were smaller than 1, the clinical significance of the observed estimates could be questioned.

The continuous measurement of body temperature during pregnancy has the potential to detect early signs of infection. Physiologically, body core temperature decreases during pregnancy due to systemic vasodilation and an increase in skin perfusion, ventilation, and plasma volume [49,50]. The effect of vasodilatation is influenced by the female sex hormones. Of these, estradiol promotes heat dissipation by vasodilation, and progesterone promotes heat conservation and higher body temperatures [2]. In contrast, skin temperature increases during pregnancy, especially at the acral area [1,49]. In our study, we measured WST, which remained relatively stable throughout pregnancy. This might be explained by the fact that we measured the temperature at night. During sleep, vasoconstriction is reduced, and skin blood flow is increased, especially at the acral area, thus increasing the skin temperature [51,52]. Our participants wore a wearable device that continuously measured wrist skin temperature during sleep. Since the first 90 and the last 30 min of recorded data were excluded in order to eradicate temperature shifts during the phase of falling asleep and awakening, a nocturnal steady state of temperature was captured. In our multivariate analysis, lower WST seemed to be related to frequent feelings of anxiety, yet the statistical model with an interaction term did not significantly improve the model fit. We, therefore, cannot state that the extent of potential changes in WST over time was dependent on the frequency of feeling anxious.

Remote monitoring during pregnancy using a wearable sensor has several potential advantages compared to the current practice: it provides continuous data on several physiological parameters at the same time while the pregnant woman is in her everyday environment. From a research point of view, it helps us to gain more information on the normal course of these parameters during pregnancy. Clinically, it has the potential to provide early signs of an abnormal pregnancy course, e.g., early signs of infection (temperature), hypertensive disorders (HRV), or mental distress/depression (HRV). Continuous remote monitoring could reduce the frequency of in-person visits. Other research projects using wearable sensors have found benefits concerning health care costs and health resource consumption in the fields of basic prenatal care, prenatal care for patients with hypertensive disease in pregnancy, and prenatal care for women with gestational diabetes. They reported good acceptance by the pregnant women [53]. The COVID-19 pandemic underlined the need for the introduction of wearable sensor technology into the prenatal care routine, especially in rural areas.

Our study has several limitations. Firstly, factors that could have potentially influenced the physiological parameters in pregnancy besides emotions were not evaluated. We chose to only monitor the pregnant women during nighttime to improve compliance and to reduce potential bias by numerous external factors, which would have decreased comparability of the measurements within our small cohort. These potential influencing factors include sexual activity, exercise, food intake, sleep duration, sleep quality, or fever/inflammation [54]. An examination of these factors on the curve patterns and their agreement would have been particularly interesting, but the size of our study group was too small to calculate their impact. This would be a valuable aspect for future studies. However, measuring physiological parameters at night might have reduced this bias to some extent. Secondly, we present data from a pilot study with a limited number of included pregnant women. Therefore, small variations can have a big impact on the pooled curves. Future studies should aim at recruiting a larger cohort. Thirdly, we only included healthy pregnant women to reflect the normal course of the basic physiological parameters.

## 6. Conclusions

Our data indicate that the sensor bracelet was feasible for monitoring vital signs during pregnancy and for establishing modern curves for the physiological course of these parameters using close or even continuous monitoring. The nightly heart rate was lower in our study compared to previously reported values, and there was no significant association with the studied emotions. We did observe a slight increase in heart rate throughout gestation, which corroborates earlier findings. Heart rate variability decreased during the first two-thirds of pregnancy followed by a slight increase in the third trimester. We found the decrease to be more pronounced in women who frequently felt stressed. The findings from our study match previous reports of stable breathing rates throughout pregnancy, with a slight decline from the second trimester onwards, yet we observed a slight increase in early pregnancy. Wrist skin temperature remained relatively stable throughout pregnancy, although others have reported an increase. This might be explained by the measurement being taken during nighttime, when the peripheral vasoconstriction is reduced leading to a higher skin temperature at the acral area. Wearables have the potential to detect early signs of pathologies during pregnancy like hypertensive disorders or infection. Our findings need to be validated by larger clinical trials using similar electronic wearables during nighttime and daytime.

## Figures and Tables

**Figure 1 sensors-23-06620-f001:**
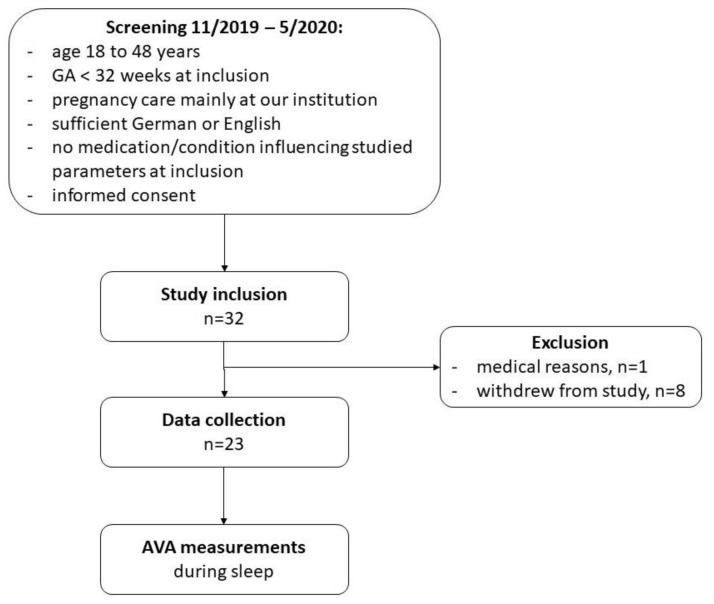
Flowchart of inclusion.

**Figure 2 sensors-23-06620-f002:**
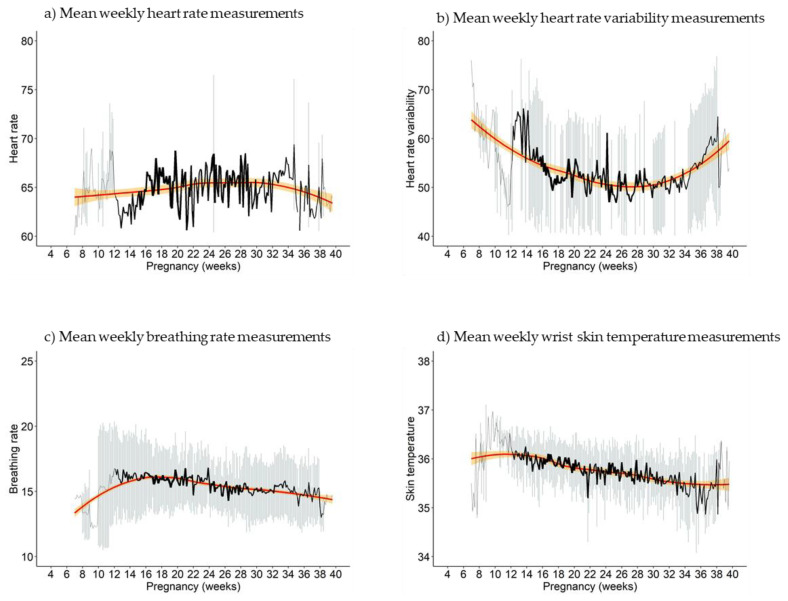
Mean weekly measurements of physiological parameters throughout pregnancy for n = 23 women. The x-axis depicts the gestational weeks, and the y-axis depicts the mean weekly measurements; error bars show the standard deviation. The red curve is a smoothed mean curve, and the orange area depicts the confidence interval of the smoothed mean curve.

**Figure 3 sensors-23-06620-f003:**
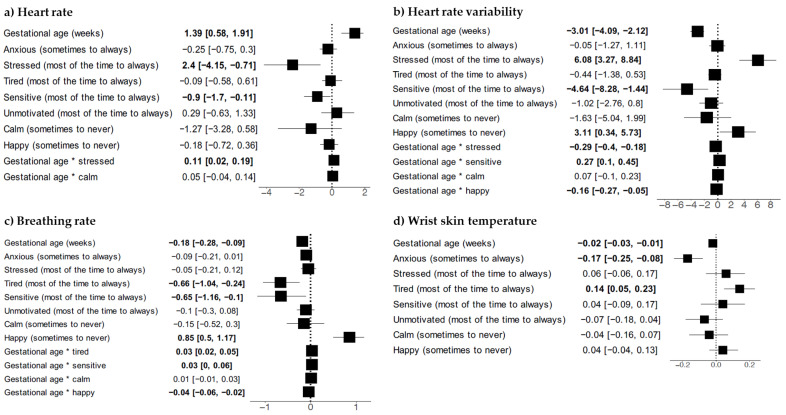
Results from multivariate linear mixed models on the effect of subjective emotions on the trajectory of four physiological parameters over the course of pregnancy. Significant values are printed in bold. * denotes the interaction between the preceding and the subsequent variable.

## Data Availability

Data available on request due to restrictions, e.g., privacy or ethical concerns.

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
