# Peer review of "Changes in Heart Rate, Heart Rate Variability, Breathing Rate, and Skin Temperature throughout Pregnancy and the Impact of Emotions—A Longitudinal Evaluation Using a Sensor Bracelet"

_sensors, 2023, doi:10.3390/s23146620_

Round 1

Reviewer 1 Report

title: too long, complicated and seems incomplete (electronic wearable – what wearable?)

line 47: HRV is just a general term. Specify that exactly which calculated parameters are affected.

line 101: again HRV is just a collecting term. The RR intervals recorded, and various parameters can be calculated. Which ones were used in the study?

figure 1: the sign for the number of observations is sometimes capitalized, sometimes not. Mainly n used, change to that everywhere in the text.

furthermore, figures and tables should appear after the first mention, not before. Correct that everywhere.

figures: they are low quality; it has to be improved.

line 193: As I explained, there is no such measures that HRV. The length of RR intervals is recorded with ECG precision, and out of that information various parameters like SD1, SD2, PNN50, SDNN, etc. can be calculated. So it has to be clarified, that what is that number exactly.

Author Response

Thank you very much for your constructive feedback. We appreciate your efforts and have made substantial changes to the manuscript as described below.

Point 1: title: too long, complicated and seems incomplete (electronic wearable – what wearable?)

Response 1: We have changed and shortened the title: Changes of heart rate, heart rate variability, breathing rate and skin temperature throughout pregnancy and the impact of emotions – a longitudinal evaluation using a sensor bracelet. 

Point 2: Heart rate variability

line 47: HRV is just a general term. Specify that exactly which calculated parameters are affected.

line 101: again, HRV is just a collecting term. The RR intervals recorded, and various parameters can be calculated. Which ones were used in the study?

line 193: As I explained, there is no such measures that HRV. The length of RR intervals is recorded with ECG precision, and out of that information various parameters like SD1, SD2, PNN50, SDNN, etc. can be calculated. So it has to be clarified, what is that number exactly.

Response 2: Thank you very much for your careful review of our article. Of course, you are absolutely right that we need to specify how the HRV was calculated. We added the following paragraph including an additional reference to the methods section: “Heart rate is the number of heart beats per minute. Heart rate variability represents the time variation in the interval between two consecutive heart beats. It can be calculated and analyzed in different ways. For this study, the standard deviation of the N-N intervals (SDNN) was used. N-N stands for “normal” beats, as abnormal or false beats, like ectopic beats not coming from the sinoatrial node, are removed from the analysis [Shaffer F, Ginsberg JP. An Overview of Heart Rate Variability Metrics and Norms. Front Public Heal 2017; 5: 258. doi:10.3389/FPUBH.2017.00258].”

Point 3: figure 1: the sign for the number of observations is sometimes capitalized, sometimes not. Mainly n used, change to that everywhere in the text.

Response 3: We changed it to n in figure one and checked it everywhere in the documents.

Point 4: furthermore, figures and tables should appear after the first mention, not before. Correct that everywhere.

Response 4: We have adjusted the position of figures and tables.

Point 5: figures: they are low quality; it has to be improved.

Response 5: We adjusted the quality of the figures. We will also provide them to the journal as separate .jpeg files, so they can be properly inserted by the editorial team.

Reviewer 2 Report

In my opinion, the article is well written, but I suggest the following improvements:

1. Change section 1 introduction to two sections, where section I continues to be Introduction and the purpose of the article is added and section 2 is Related Works and Contributions

2. In section 2 of the original Material and Methods, indicate the alternative research methods and justify the chosen research method and especially the type of data acquisition selected.

3. The images in figure 2 see the possibility of improving their resolution.

4. In section 5 the conclusions presented are very generic. Please present the conclusions quantitatively, based on the results obtained and also indicate their validation.

Author Response

Thank you for your supportive comments. We have carefully revised our manuscript as recommended by the reviewers. Please see the details below.

Point 1: Change section 1 introduction to two sections, where section I continues to be Introduction and the purpose of the article is added and section 2 is Related Works and Contributions

Response 1: We divided the introduction as suggested and added a section on Related Works and Contributions. Please see the updated manuscript for the updated section.

Point 2: In section 2 of the original Material and Methods, indicate the alternative research methods and justify the chosen research method and especially the type of data acquisition selected.

Response 2: We have added the following paragraph to the methods section: “We chose the Ava sensor bracelet for our study as it is non-invasive, discrete, and convenient to wear. Furthermore, it is a CE approved medical device in Europe and has been proven feasible in other clinical settings. Continuous measurements are more robust than punctual measurements and are potentially able to detect pathologies at an earlier stage in pregnancy. The concept of measuring the physiological parameters during sleep excludes various potential interfering factors.”

Point 3: The images in figure 2 see the possibility of improving their resolution.

Response 3: We adjusted the quality of the figures. We will also provide them to the journal as separate .jpeg files, so they can be properly inserted by the editorial team.

Point 4: In section 5 the conclusions presented are very generic. Please present the conclusions quantitatively, based on the results obtained and also indicate their validation.

Response 4: We changed the conclusion section as follows: “Our data indicates that the sensor bracelet was feasible to monitor vital signs in pregnancy and to establish modern curves for the physiological course of these parameters using close or even continuous monitoring. The nightly heart rate was lower in our study compared to previously reported values and there was no significant association with the studied emotions. We did observe a slight increase in heart rate throughout gestation, which corroborates earlier findings. Heart rate variability decreased during the first two thirds of pregnancy followed by a slight increase in the third trimester. We found the decrease to be more pronounced in women who frequently felt stressed. The findings from our study match previous reports of stable breathing rates throughout pregnancy, with a slight decline from the second trimester onwards, yet we observed a slight increase in early pregnancy.  Wrist skin temperature remained relatively stable throughout pregnancy although others have reported an increase. This might be explained by the measurement during night time, when the peripheral vasoconstriction is reduced leading to a higher skin temperature at the acra. Wearables have the potential to detect early signs of pathologies during pregnancy like hypertensive disorders or infection. Our findings need to be validated by larger clinical trials using similar electronic wearables during night- and day time.“

Reviewer 3 Report

This article used a fitness bracelet to measure and assess several health parameters in healthy pregnant women continuously, including skin temperature, heart rate, heart rate variability, and breathing rate. It also assessed, if maternal emotions can influence the measured parameters. This study has important significance to help us find early signs of an abnormal pregnancy course. However, there are still some problems in this article that need to be answered and revised. Some comments are listed below.

1. In “2.1 Sample population”, why is “gestational age below 32 weeks at inclusion” used as the inclusion criteria when including pregnant women? After all, the pregnant women participating in the study had a gestational age range of 8 to 25 weeks.

2. This article emphasized the importance of continuous monitoring of physiological parameters of pregnant women, but only the physiological parameters of pregnant women during night rest were monitored in the study. Why cannot the physiological parameters of pregnant women be monitored throughout the day?

3. In “2.2 Data acquisition”, to stabilize the physiological parameters, at least four hours of relatively uninterrupted sleep each night is required. And, the first 90 and the last 30 minutes of each night’s data were excluded to avoid disturbances of the falling-asleep and wake-up phases. If a pregnant woman suddenly wakes up during sleep, will it have an impact on the monitored physiological parameter data? Has this study considered this issue?

4. In “1. Introduction”, there are reports indicating that skin temperature is rising with the ongoing time of pregnancy due to continuous vasodilation. However, in this study, the skin temperature of pregnant women decreased with the increase of pregnancy. Why is this?

5. The order of the data figures in Figure 2 and the text introduction in “3. Results” for different physiological parameters is different, which can cause confusion for readers when reading. Please make both in the same order.

6. Please correspond the different physiological parameter data figures in Figure 2 in the text one by one, just like Figure 1 marked in the text.

7. What are the cohort characteristics listed in Table 1 to illustrate and for what purpose?

8. What do the different color curves in the data figures of the different physiological parameters represent for in Figure 2? Please explain clearly in the caption.

9. For the effect of subjective emotions on the trajectories of four physiological parameters over the course of pregnancy, can it be visually displayed in the main text in the form of figures or tables, rather than placing data in supporting materials.

None

Author Response

We thank Reviewer 3 for the detailed feedback, which helped us to improve the quality of our manuscript. Please see the details below.

Point 1: In “2.1 Sample population”, why is “gestational age below 32 weeks at inclusion” used as the inclusion criteria when including pregnant women? After all, the pregnant women participating in the study had a gestational age range of 8 to 25 weeks.

Response 1: We excluded pregnant women >32 weeks of gestation from the beginning of the study right away during the screening phase. We chose an upper limit for gestational age at inclusion to be able to continuously monitor basic physiological parameters over a longer course of pregnancy as we intended to create modern standard curves using an electronic wearable. If we had included women with 32 weeks or more, the remaining time of pregnancy would not have allowed us to monitor a relevant episode.

We added the following sentence to the methods section to clarify this point: “We chose the upper limit of 32 weeks for gestational age at inclusion to be able to continuously monitor basic physiological parameters over a longer course of time during pregnancy as we intended to create modern standard curves using an electronic sensor bracelet.”

Point 2: This article emphasized the importance of continuous monitoring of physiological parameters of pregnant women, but only the physiological parameters of pregnant women during night rest were monitored in the study. Why cannot the physiological parameters of pregnant women be monitored throughout the day?

Response 2: Thank you for addressing this aspect. There were several reasons which led to the decision to just include nightly measurements: Firstly, we believe that it would not have been acceptable for some women to wear the sensor bracelet 24 hours per day for many months and would potentially have decreased compliance and willingness to participate in the study. Secondly, we think that measurements during sleep are more comparable to each other as they are not influenced by physical activity as well as changing environments. But of course, a 24-hour measurement over a longer time period would be an interesting research question for future clinical trials.

We have added some statements concerning this point to the limitations in the discussion section: “Our study has several limitations. Firstly, factors that could have potentially influenced the physiologic parameters in pregnancy besides emotions were not evaluated. We chose to only monitor the pregnant women during night time to improve compliance and to re-duce potential bias by numerous external factors, which would have decreased comparability of the measurements within our small cohort. These potential influencing factors include sexual activity, exercise, food intake, sleep duration, sleep quality or fever/inflammation. An examination of these factors on the curve patterns and their agreement would have been particularly interesting but the size of our study group was too small to calculate their impact. This would be a valuable aspect for future studies.” 

Point 3: In “2.2 Data acquisition”, to stabilize the physiological parameters, at least four hours of relatively uninterrupted sleep each night is required. And, the first 90 and the last 30 minutes of each night’s data were excluded to avoid disturbances of the falling-asleep and wake-up phases. If a pregnant woman suddenly wakes up during sleep, will it have an impact on the monitored physiological parameter data? Has this study considered this issue?

Response 3: The sensor bracelet also includes a sleep pacer. All measurements during night time, when the woman was awake and not sleeping, were not recorded. For the analysis, the bracelet needs at least four hours of sleep excluding falling asleep and waking up. This technique was chosen to reduce the influencing factors mentioned under Point 2. Therefore, this study did not evaluate the influence of factors like physical activity on the physiological parameters as mentioned in the discussion section. This would be an interesting aspect for future trials.

We added this information to the methods section.

Point 4: In “1. Introduction”, there are reports indicating that skin temperature is rising with the ongoing time of pregnancy due to continuous vasodilation. However, in this study, the skin temperature of pregnant women decreased with the increase of pregnancy. Why is this?

Response 4: We tried to clarify this point in the discussion section:

„Physiologically, body core temperature decreases during pregnancy by systemic vasodilation and an increase in skin perfusion, ventilation, and plasma volume (…). In contrast, skin temperature increases during pregnancy, especially at the acra. In our study, we measured WST, which remained relatively stable throughout pregnancy. This might be explained by the fact that we measured the temperature at night. During sleep, vasoconstriction is reduced, and skin blood flow is increased, especially at the acra, thus increasing the skin temperature. Our participants wore a sensor bracelet that continuously measured wrist skin temperature during sleep. Since the first 90 and the last 30 minutes of recorded data were excluded in order to eradicate temperature shifts during the phase of falling asleep and awakening, a nocturnal steady state of temperature was captured.“

Point 5: The order of the data figures in Figure 2 and the text introduction in “3. Results” for different physiological parameters is different, which can cause confusion for readers when reading. Please make both in the same order.

Response 5: We changed the order within Figure 2 a-d), so it is the same as in the text.

Point 6: Please correspond the different physiological parameter data figures in Figure 2 in the text one by one, just like Figure 1 marked in the text.

Response 6: We added the links to Figure 2 a-d) in the text of the results section.

Point 7: What are the cohort characteristics listed in Table 1 to illustrate and for what purpose?

Response 7: You are absolutely right that these informations are not necessary for our research question. We have removed Table 1 and added information about mean age, mean BMI and parity to the text of the results section.  

Point 8: What do the different color curves in the data figures of the different physiological parameters represent for in Figure 2? Please explain clearly in the caption.

Response 8: We added the following explanation to the caption of figure 2: “The red curve is a smoothed mean curve and the orange area depicts the confidence interval of the smoothed mean curve.”

Point 9: For the effect of subjective emotions on the trajectories of four physiological parameters over the course of pregnancy, can it be visually displayed in the main text in the form of figures or tables, rather than placing data in supporting materials.

Response 9: We have added figure 3 to the manuscript, which depicts the multivariate analysis of the influence of emotions on the four physiological parameters. We have replaced the table for the bivariate model in the supplementary material section by figures as well.

Round 2

Reviewer 3 Report

After the revision, the article can be accepted.

None